# The Impact of Maternal Chronic Inflammatory Conditions on Breast Milk Composition: Possible Influence on Offspring Metabolic Programming

**DOI:** 10.3390/nu17030387

**Published:** 2025-01-22

**Authors:** Gabriela Arenas, María José Barrera, Susana Contreras-Duarte

**Affiliations:** 1Facultad de Medicina y Ciencia, Universidad San Sebastián, Santiago 7510602, Chile; garenasu@correo.uss.cl; 2Facultad de Odontología y Ciencias de la Rehabilitación, Universidad San Sebastián, Santiago 7510157, Chile; maria.barrera@uss.cl; 3Facultad de Ciencias para el Cuidado de la Salud, Universidad San Sebastián, Santiago 8420524, Chile

**Keywords:** breast milk, inflammation, obesity, diabetes, hypercholesterolemia, hypertension, metabolic syndrome

## Abstract

Breastfeeding is the best way to provide newborns with crucial nutrients and produce a unique bond between mother and child. Breast milk is rich in nutritious and non-nutritive bioactive components, such as immune cells, cytokines, chemokines, immunoglobulins, hormones, fatty acids, and other constituents. Maternal effects during gestation and lactation can alter these components, influencing offspring outcomes. Chronic inflammatory maternal conditions, such as obesity, diabetes, and hypertension, impact breast milk composition. Breast milk from obese mothers exhibits changes in fat content, cytokine levels, and hormonal concentrations, potentially affecting infant growth and health. Similarly, diabetes alters the composition of breast milk, impacting immune factors and metabolic markers. Other pro-inflammatory conditions, such as dyslipidemia and metabolic syndrome, have been barely studied. Thus, maternal obesity, diabetes, and altered tension parameters have been described as modifying the composition of breast milk in its macronutrients and other important biomolecules, likely affecting the offspring’s weight. This review emphasizes the impact of chronic inflammatory conditions on breast milk composition and its potential implications for offspring development through the revision of full-access original articles.

## 1. Introduction

Breastfeeding provides essential nutrients and fosters a unique bond in the dyad [1]. Breast milk is rich in bioactive components, including immune cells, hormones, and fatty acids, which support infant growth and development [2,3,4,5,6,7]. However, maternal health conditions, such as obesity, diabetes, and hypertension, are described to modify the composition of breast milk [8,9,10,11,12,13]. These chronic inflammatory states alter levels of macronutrients, cytokines, and hormones, impacting offspring’s weight [14,15,16]. This article examines the effects of maternal inflammatory conditions on breast milk composition and explores their implications for child development.

## 2. Methodology

In this review, we aimed to show that metabolic diseases such as obesity, diabetes, hypertension, dyslipidemia, and metabolic syndrome, which have joint systemic inflammation, can impact women during gestation and continue during lactation, affecting breast milk quality and composition. In addition, this review briefly describes the scarce existing information regarding the effects of these alterations on the offspring later in life. For this, we performed a comprehensive literature review using Science Direct, PubMed, and Medline (https://www.ncbi.nlm.nih.gov/pubmed) search for articles published up to December 2024. For this purpose, the search was performed using the following mesh terms and combinations of words was done to search: (“breast milk” OR “breastmilk” OR “breastfeeding” OR “lactation” OR “lactation programming” OR “maternal milk” OR “human milk” OR breast milk composition) AND (“chronic inflammation” OR “obesity” OR “overweight” OR “diabetes” OR “diabetes mellitus type 2” OR “ type 1 diabetes” OR “insulin resistance” OR “preeclampsia” OR “hypertension” OR “hypertensive syndrome” OR “hypercholesterolemia” OR “dyslipidemia” OR “maternal dyslipidemia” OR “maternal inflammation” OR “metabolic syndrome” OR “MetS” OR “epigenetics” OR “fetal programming”) AND (“postpartum” OR “puerperium” OR “immunity” OR “immune system” OR “immune cells” OR “macronutrients” OR “offspring”). We selected studies in humans published as full-length articles. The electronic search was integrated by manually searching the reference lists of the selected publications and reviews on the issue to identify any other relevant publications.

Inclusion and exclusion criteria studies were included if they provided quantitative information on the relationship between previously mentioned variables and if the articles were written in English idiom.

## 3. Breastfeeding, Breast Milk Composition, the Impact of Geographic Regions, and Maternal Dietary Habits on Breast Milk Composition

Breastfeeding is essential to the neonatal and infant periods [1]. It is widely known as the best way to provide newborns with crucial nutrients and produce a unique bond between mother and child [1,17]. Compared with alternative formulas, maternal milk is considered the preferred source of nutrition for newborns [17]. In this regard, the World Health Organization (WHO) recommends exclusively breastfeeding for the first six months of an infant’s life to receive the optimal benefits of breast milk [18]. Under normal conditions, breastfeeding is associated with a lower incidence of globally significant infectious diseases like diarrhea and pneumonia and a decreased prevalence of critical chronic diseases, such as obesity and diabetes [19].

Breast milk is a complex biological fluid secreted by alveolar epithelial cells from the mammary gland, which provides all the necessary nutrient requirements for neonatal growth and maturation [20]. Its composition is rich in nutritional and non-nutritional bioactive components, including enzymes, hormones, and immune factors such as chemokines, cytokines, immunoglobulins, and immune cells [2]. Breast milk contains irreplaceable components essential for optimal postnatal development, unlike most infant formulas, which lack bioactive and immune factors [21].

Breast milk composition is dynamic and may vary in response to the infant’s nutritional needs depending on the nursing stage, maternal diet, and many other environmental factors [2,3,4,5,6,7]. This ability to modify its composition according to the environment suggests that it can also change in maternal pathological states, especially those that alter the balance in the immune response, favoring inflammation in chronic conditions [22]. Research in human and animal models indicates that maternal exposures can alter specific bioactive components in breast milk, influencing offspring outcomes [2]. Thus, recent studies have pointed out that breast milk is a biological nutrient system [23] that can be disturbed under certain maternal pro-inflammatory conditions, being conditionally perfect [24]. In this regard, breast milk accomplishes its basic nurturing function and, at the same time, executes a second and important action, which is infant oral programming through the transmission of factors that modify some epigenetic mechanisms that can contribute to the epigenetic programming in the nursling [25].

In normal conditions, breast milk is made up of about 87–88% water and solid components, including about 7% carbohydrates, 1% proteins, and 3.8% fat [26]. Among the constituents of breast milk, the fat fraction provides 44% of the total energy intake in infants [27]. It is secreted in small droplets that contain lipids (milk fat globule), which are a source of essential fatty acids (FAs) that are crucial for the neurological and immune development of a child [28,29,30]. Most of them are saturated FAs and monounsaturated FAs, followed in quantity by omega-3 (*n*-3) or omega-6 (*n*-6) polyunsaturated FAs (PUFAs) [21,31]. *n*-3 PUFAs decrease the production of inflammatory cytokines, while *n*-6 increase intestinal inflammatory responses and oxidative stress [32]. The ideal *n*-6:*n*-3 PUFA ratio should range between 1:1 to 5:1 to maintain a healthy balance and prevent chronic diseases [33]. An increased ratio is associated with pro-inflammatory effects [26]. Breast milk is rich in immune cells, which migrate to the breast via the lymphatic vessels and systemic circulation, playing a crucial role in developing the infant’s immune system [34,35]. Breast milk leukocytes are principally composed of neutrophils (40–65%), followed by macrophages (35–55%) and lymphocytes T and B (5–10%) [34,36,37,38]. These immune cells produce and release cytokines, such as interleukin 1β (IL-1β), IL-6, IL-10, IL-13, tumor necrosis factor-alpha (TNF-α), interferon-gamma (IFN-γ), transforming growth factor beta (TGF-β), among others [34,39]. Some of these, like IL-6 and TNF-α, have pro-inflammatory properties and, under normal conditions, offer beneficial effects in the defense against gastrointestinal and respiratory diseases [34,40,41]. Meanwhile, IL-10 and TGF-β have anti-inflammatory effects, promoting B cell differentiation and playing a regulatory role in the neonatal gastrointestinal tract [34,40,41,42]. The abundance of leukocytes and cytokines in breast milk is highly dynamic, changing throughout lactation and being influenced by the maternal and infant health status [34,41,43,44].

Maternal dietary habits play a crucial role in determining the composition of breast milk, influencing various bioactive components. Notably, the concentrations of fat, fatty acids, and water-soluble vitamins, such as vitamins A, C, B6, and B12, can vary in human milk due to maternal nutrition [45,46,47,48]. Among these bioactive components, lipids are likely the most affected by maternal diet. Studies have shown that a maternal high-fat and high-lipid diet can increase milk fat content by approximately 12–25% compared to a low-fat diet [49,50,51,52]. Ward et al. determine the acute effect of increased sugar (150.4 g sugar/day) and fat (129.7 g fat/day) consumption on breast milk lipids, finding that triglycerides significantly increased in both diets, with a greater response to the higher sugar intake compared to the control diet (mean difference of 3.05 g/dL ± 0.39 and 13.8 g/dL ± 0.39, respectively; *p* < 0.001) [53]. Breast milk cholesterol concentration increased only with the high-sugar diet (0.07 g/dL ± 0.005 vs. 0.04 g/dL, *p* < 0.005) [53]. These intervention studies provide evidence that maternal diet can impact total milk lipids.

Breast milk fatty acids are derived from endogenous synthesis in the mammary gland and uptake from maternal plasma, with both processes being influenced by maternal nutrition [45]. Dietary fatty acids are rapidly transferred into milk, and the composition of human milk fat can change to reflect the dietary fat profile [49]. In this regard, Kim et al. found positive correlations between dietary intake of eicosapentaenoic acid (EPA), docosahexaenoic acid (DHA), *n*-3 fatty acids, *n*-6 fatty acids, saturated fatty acids (SFA), and PUFAs and their concentrations in the breast milk of South Korean mothers [54]. Similar correlations were observed in lactating women in China, where the maternal dietary fatty acid composition was positively associated with SFA, PUFA, and DHA concentrations (*p* < 0.001) and negatively with monounsaturated fatty acids (MUFA) (*p* > 0.05) in breast milk [55]. High-fat diets can increase omega-6 fatty acids in breast milk, which may exacerbate inflammatory responses [54,56,57,58]. Conversely, a diet supplemented with omega-3 fatty acids can enhance DHA and EPA concentrations in breast milk, potentially mitigating the risks associated with chronic inflammation [59,60,61,62].

The link between maternal diet and chronic inflammatory conditions is well-established. Poor dietary choices can lead to metabolic dysregulation, contributing to conditions such as obesity, diabetes, and hypercholesterolemia, which not only affects maternal health but also alters the bioactive components of breast milk [63,64,65,66,67]. Therefore, addressing maternal dietary habits is essential, particularly in the context of chronic inflammatory conditions. A balanced, nutrient-rich diet can help mitigate some of the negative effects these conditions have on breast milk composition, thereby supporting better health outcomes for both mother and infant. Incorporating dietary counseling into postpartum care could enhance the bioactive profile of breast milk, promoting its role as a critical component in the early-life prevention of inflammation-related diseases.

Breast milk composition varies dynamically across different ethnicities and geographic regions [68]. Meta-regression analyses indicate that the study region significantly contributes to the variation in lipid profiles [68]. Kumar et al. investigated the influence of geographical location on breast milk lipid composition, focusing on Europe (Spain and Finland), Africa (South Africa), and Asia (China) [69]. The study found significant differences in total PUFAs (*p* < 0.001), *n*-6 PUFAs (*p* < 0.001), *n*-3 PUFAs (*p* = 0.003), and SFA (*p* < 0.001) across countries. Chinese women were observed to have the highest levels of *n*-6 PUFAs (25.6%) and lower levels of SFA (34.7%) compared to other regions. Additionally, Chinese and Finnish samples exhibited relatively higher levels of *n*-3 PUFAs [69]. Western countries tend to have higher concentrations of MUFAs and omega-9 (18:1*n*-9) but lower levels of PUFAs, including 18:2*n*-6, 20:4*n*-6, 18:3*n*-3, 20:5*n*-3, 22:6*n*-3, and total *n*-6 PUFA in human milk compared to non-Western countries (*p* < 0.001) [68]. In contrast, no significant regional difference was detected for SFA (*p* = 0.799) and most other subtypes of fatty acids [68]. Yuhas et al. compared the fatty acid contents of human milk across nine countries (Australia, Canada, Chile, China, Japan, Mexico, Philippines, the United Kingdom, and the United States), finding that the proportion of saturated fatty acids was relatively constant among these countries, except the Philippines, where levels of lauric and myristic acids were elevated [68,70]. Arachidonic acid (ARA) concentrations range from 0.36% in the United Kingdom to 0.49% in China, and DHA levels range from 0.17% to 0.99%, with the highest levels in Japanese milk and the lowest levels in Canadian and U.S. simples, highlighting the variability in human milk fatty acid content across regions [68,70]. These variations are not only attributed to geographic and dietary factors but also to lactation stages, time of day, and sample collection year [68]. Understanding these differences is crucial for nutritional interventions and guidelines to optimize infant health globally.

Unlike lipids, the variation in miRNA composition across different geographic regions and populations has not been extensively studied. While such regional differences have been documented in other biofluids, such as blood and saliva, similar research on breast milk remains limited. Given the critical role of miRNAs in regulating immune and metabolic processes in infants, understanding their regional variation could provide valuable insights into how environmental, dietary, and genetic factors influence breast milk composition. Investigating these variations is essential to identify potential disparities in the health benefits of breast milk, which could inform the development of targeted nutritional interventions to optimize infant health across diverse populations.

In this review, we highlight the effects of maternal pathologies characterized by chronic inflammation on breast milk composition and the implications of this on offspring’s development later in life. Understanding the influence of maternal health status on breast milk components is crucial for developing nutritional strategies that optimize the benefits derived from breastfeeding.

## 4. Chronic Inflammation

Inflammation is a biological response of the immune system that can be triggered by various factors, including pathogens, damaged cells, and toxic compounds [71]. During inflammation, immune cells are recruited from circulation, infiltrate the tissue, differentiate, and produce inflammatory mediators as cytokines to achieve healing and restore the tissue’s homeostasis [72]. Acute inflammation can be triggered in minutes to hours and lasts hours to days [73]. On the other hand, chronic inflammation continues to acute inflammation when the tissue injury or infection cannot be resolved. Chronic inflammation recruits monocytes; however, it also usually involves the recruitment and activation of adaptive immune cells such as lymphocytes [74,75]. Interestingly, chronic inflammatory conditions are not caused by traditional triggers but are mainly caused by tissue malfunction [76]. The inflammation that characterizes most chronic diseases suggests a central role of the adaptive and innate immune system in the risk and pathogenesis of these conditions [77].

The newborn’s immune system is influenced by maternal immunity transferred via the placenta and breast milk [78]. Breast milk contains components that can modulate the immune system of infants by providing protection and facilitating development, tolerance, and an appropriate inflammatory response [32,35]. Researchers indicate that chronic inflammatory diseases in adulthood seem to be associated with early prenatal and postnatal malnutrition [77,79]. Inflammation early in life may lead to adverse neurodevelopmental outcomes, highlighting the importance of mitigating inflammation during the newborn period [32]. It is essential to understand the influence of the environment on the development of immune and metabolic programming to identify critical factors that modulate maternal and offspring health.

## 5. Chronic Inflammatory Conditions in the Mother, Their Impact on Breast Milk Composition, and Possible Role in Nursling Programming

### 5.1. Obesity

According to the WHO, obesity has emerged as a global epidemic, affecting over 650 million people worldwide [80]. The standard measure for identifying obesity is a body mass index (BMI) of 30 kg/m^2^ or higher [81]. This can be complemented by measuring waist circumference (88 cm or higher for females) to assess visceral adiposity and predict disease risk [82]. Obesity is characterized by low-grade chronic inflammation that causes multiple metabolic diseases, such as diabetes, hypertension, and dyslipidemia, among others [83].

The prevalence of obesity in women of childbearing age has been persistently increasing for the past three decades [84,85]. In 2014, the global estimation of pregnant women with obesity reached 14.6 million, and approximately one-third of these women remain obese in the postpartum period when lactation occurs [85,86]. Maternal obesity may lead to gestational diabetes mellitus (GDM), hypertension, preeclampsia, complications during delivery, and difficult breastfeeding [87]. In addition to the health concerns for mothers with obesity, children born to obese mothers are 3–5 times more susceptible to developing obesity later in life [88]. These data indicate that obesity is a persistent concern during the postpartum period and may have potential implications for maternal and infant health later in life.

Maternal obesity is related to excess adipose mass, which affects hormonal and some vitamin [89] regulation and alters breast milk’s immunological and bioactive factors [89,90]. In this regard, breast milk of obese mothers contains higher levels of total fat, glucose, lactoferrin, C-reactive protein (CRP), and hormones such as insulin, leptin, ghrelin, adiponectin, and obestatin [8,9,10,11,12,13]. These alterations in breast milk could have implications for the infant’s metabolic development and weight regulation, e.g., they could influence the infant’s feeding behavior. Additionally, obesity has an impact on immunoglobulin (Ig) concentration, showing elevated levels of soluble IgA in colostrum (the first form of milk) [91]. In contrast, the concentrations of IgG and IgM remained unaffected by maternal BMI [91]. This suggests that certain components of the immune protection provided by breast milk, particularly those related to the transfer of immunity against pathogens, remain stable regardless of maternal obesity. Breast milk of obese mothers shows an increased ratio of pro-inflammatory omega-6 to omega-3 polyunsaturated fatty acids (PUFAs), which may influence the offspring’s cardiometabolic status and neurodevelopmental outcomes [11,12]. These changes in the profile of breast milk fatty acids are associated with alterations in insulin, CRP, leptin, adiponectin, ghrelin, interleukin (IL)-6, and tumor necrosis factor-alpha (TNF-α) [11,90]. These alterations may influence the composition of bioactive compounds delivered to the infant. However, the direct impact of these changes on neonatal health and development has not yet been thoroughly studied.

The cytokine levels in breast milk of obese mothers show conflicting results. While some studies report changes in their concentration, others do not [12,13,92,93]. This discrepancy might be due to the heterogeneous groups in these studies. However, lower levels of breast milk transforming growth factor-beta 2 (TGF-β2) and IL-6 are related to higher BMI, which may impact the maturation of the offspring’s immune system [44,94]. Some of these cytokines are derived from immune cells of the mammary gland or breast milk, while others might be transferred from maternal circulation [39]. A cross-sectional study reported phenotypic alterations of leukocyte subpopulations, such as decreased B lymphocyte fraction, between peripheral blood and colostrum from women with obesity [95]. However, there is limited information about the phenotype or status of these immune cells and, therefore, the origin of breast milk inflammation in obese mothers. Research in mice and humans with obesity demonstrates that the adipocytes, which also secrete pro-inflammatory mediators, are dysfunctional and hypertrophied, leading to tissue inflammation [96]. However, further studies are needed to explore the contribution of adipocytes in breast milk alterations in obese mothers (Figure 1).

Due to the permeability of the young infant gut, these altered immunological and bioactive compounds in breast milk may induce systemic effects by entering the offspring’s circulation [93]. Most studies have shown that pre- and postnatal nutritional excess interact with each other to exert additive effects on programming [97]. To disclose the specific role of breastfeeding, some studies have employed a cross-fostering murine model to compare the differences in pups from obese and lean dams during lactation, showing the importance of the lactation period to the later metabolic alterations in the offspring [14,98]. In mice, a maternal high-fat diet-induced redistribution of fatty acyl residues in the dam’s circulation, which was associated with increased levels of omega-6 PUFAs in both milk and the pup nursing neonate circulation [99]. High amounts of omega-6 PUFAs have been related to modifications in the eicosanoids cascade and disturbances in the immune system [99]. This may have harmful programming effects on the offspring, predisposing them to inflammatory-related conditions, such as obesity, diabetes, and hypertension, among others in mice models [14,15,16].

Clinical studies reported that elevated levels of leptin and omega-6 to omega-3 PUFAs in the human breast milk of obese mothers are associated with excessive weight gain in infants, which may lead to the development of metabolic and inflammatory conditions later in life [14,91,100,101,102,103]. Breast milk from obese mothers also has higher adiponectin levels, indicating a possible role in attenuating inflammatory processes [8]. Little is known about the impact of immune cells and cytokines in breast milk from obese mothers on offspring health; however, they are essential for immune system development and regulation of inflammatory processes in newborns [40]. These findings suggest the impact of maternal obesity and inflammation on breast milk composition, with potential implications for infant growth and development later in life, highlighting the critical need for future research to improve maternal breast milk quality (Figure 1).

### 5.2. Diabetes

Diabetes mellitus (DM) is a metabolic disease that results in elevated blood glucose levels [104]. The main subtypes of DM are Type 1 (T1DM) and Type 2 (T2DM), which classically result from defective insulin secretion (T1DM) and/or action (T2DM) [104,105]. DM in mothers may be pregestational (PGDM), consisting of T1DM and T2DM, or appear during pregnancy, known as gestational diabetes mellitus (GDM) [106]. According to the International Diabetes Federation, GDM affects approximately 14% of pregnancies worldwide, representing 20 million births annually [107]. Approximately 14.9 million women have PGDM, representing 1–2% of all pregnancies [108]. GDM and PGDM increase the risk of various complications, including preeclampsia, cardiovascular disease, and obesity [109]. These factors contribute to an increased susceptibility to the development of T2DM in both the mother and the infant [109].

Some studies suggest that the composition of breast milk from women with DM may exhibit variations compared to non-diabetic women. Colostrum of GDM mothers had a higher content of carbohydrates and energy content [110,111]. Meanwhile, calorie, fat, and protein composition were not altered [110,111]. Colostrum of GDM women had different levels of cytokines and chemokines, with increased concentrations of IL-15, IL-6, and interferon-gamma (IFN-γ) and reduced levels of granulocyte-macrophage colony-stimulating factor (GM-CSF), causing altered immune composition [112]. GDM also alters the concentration of human milk hormones, increasing insulin levels and decreasing levels of ghrelin and adiponectin; however, these relationships are limited to the early lactation stage [106,113]. Breast milk composition of GDM women also has increased lactoferrin levels [114]. Glucose was higher, and long-chain PUFAs were lower in the breast milk of PGDM women, which may result from altered fatty acid metabolism [115,116]. PGDM women have altered the mammary gland’s lipid metabolism and increased glucose and sodium levels in mature milk [117], which could affect the infant’s electrolyte balance and energy metabolism. The implications of these changes on infant health require further investigation, but they suggest that maternal DM could potentially affect the infant’s metabolic and immune programming. Also, early neonatal intake of breast milk from diabetic mothers was associated with a higher body weight in early childhood, increasing the risk of becoming overweight later in life [118] (Figure 1). The precise impacts of these alterations on the infant’s long-term health are not fully elucidated. Further research is necessary to discern how maternal diabetes may affect the composition of breast milk and whether these changes have significant consequences for the child’s development and health.

### 5.3. Hypercholesterolemia

Hypercholesterolemia is a lipid metabolism disorder characterized by elevated cholesterol levels in the blood, mainly low-density lipoprotein cholesterol (LDL-C), strongly associated with cardiovascular disease [119]. According to the WHO, approximately 39% of adults worldwide have elevated cholesterol levels, with a relatively higher prevalence among women (40%) compared to men (37%) [120]. As described in other pathologies reviewed earlier, breastfeeding is a protective factor to prevent this lipidic alteration in the dyad. Specifically, breastfeeding for more than two years decreases the 16% probability of having LDL-related disorders among women aged 30–39 years compared to women who did not breastfeed [121]. For breastfed adults, it is also described that it may have benefits related to reduced risk of developing atherosclerosis [122]. Breastfeeding for over three months is associated with improved maternal cholesterol profiles at one year postpartum [123,124]. Lactation functions as a physiological pathway for the excretion of triglycerides and cholesterol, which helps reduce maternal serum concentrations during this period [125,126]. However, these lipid levels may return to baseline after lactation ends [125,126]. This lipid-lowering effect postpartum mediated by lactation may contribute to a reduced risk of developing cardiovascular diseases later in life in women who breastfed. Therefore, breastfeeding not only benefits the infant but may also have significant protective effects on maternal cardiovascular health by improving lipid metabolism and reducing the likelihood of persistent dyslipidemia.

Maternal dyslipidemia in pregnancy is associated with adverse pregnancy outcomes for both mother and fetus, including preeclampsia and an increased risk of gestational diabetes [127]. Previous studies have shown that the prevalence of postpartum lipid alteration ranges from 38.5% to 52%, depending on the observation period after delivery [128,129,130]. Regardless, when dyslipidemia or hypercholesterolemia occurs during breastfeeding or the postpartum period, these disorders have not been articulated as clinically significant problems. As such, lipids are not determined during this period as public health policy. Even though several articles mention this alteration as a significant problem during gestation with important consequences for the dyad regarding their cardiovascular health [131,132,133,134,135], to our knowledge, there are no studies that have researched the development of hypercholesterolemia during puerperium or postpartum period as such, the impact on maternal and infant cardiovascular health, or the possible impact of this lipidic disorder in breast milk composition or quality that these women produce.

All the available information described in the literature are observational reports regarding the variation of lipids before pregnancy (total cholesterol 160–210 mg/dL), during the first trimester of pregnancy (total cholesterol 167–198 mg/dL), second trimester (total cholesterol 200–279 mg/dL), third trimester (total cholesterol 239–317 mg/dL), at delivery (total cholesterol 240–312 mg/dL), and as time goes by in the postpartum period, lipid levels normalize in normal gestation [132,136,137,138,139,140]. However, there is another group that, according to the literature, shows hypercholesterolemia during pregnancy and the postpartum period. Variations in maternal lipids levels have also been reported due to physiologic changes in the menstrual cycle [141], specific maternal pathologies [128,142,143,144,145,146,147,148,149,150,151] such as depression [143,147,152], alterations in glucose metabolism [145,146,148,151], and tension alterations [145,149,150]. In addition, lipids during postpartum also were evaluated under dietary and exercise interventions [153,154], weight loss or retention [153,155,156,157,158,159], parity [160,161], lactation versus no lactating [123,162], type of lactation [163], extension of lactation [126,164,165], among others, all of them are summarized in the following table sorted by year of publication (Table 1).

Interestingly, what it is relevant to highlight from these studies is that women who had lipid levels higher than 300 mg/dL during the third trimester of gestation remained with their lipid levels high for a more extended period during the postpartum period, in comparison to those women who during the third trimester of pregnancy had their lipid levels lower to 300 mg/dL [132]. Additionally, it was observed, related to parity [136,137,160], that multiparous women presented higher lipid levels than nulliparous women [160]. Worryingly, these women presented lower levels of HDL cholesterol and higher levels of LDL cholesterol during the postpartum period [139], suggesting that pregnancy increases their vascular risk, likely sensitive to diet consumption. Even though the authors observed this alteration during the postpartum period, this evidence was not subject to further analysis.

Most studies centered on lipid levels during the postpartum period mention that lipid concentration decreases to “normal levels”, i.e., to 200 mg/dL in a year [136,137,140]. Jimenez et al., 1988 described that the first 40 days postpartum decrease unesterified cholesterol levels [138], which is relevant since free cholesterol in circulation is biologically active and has cytotoxic effects [171]. However, this study did not consider whether women were breastfeeding and how these lipid levels might be affected by this parameter. In this respect, TC and TG levels decrease during the postpartum period, although TC and TG from breastfeeding women decrease their levels faster than those who do not breastfeed [172]. However, no studies mention maternal lipidic alterations, such as hypercholesterolemia, that could be present during the lactation period as a pathology or that this lipid change could be a physiological response to lactation demands (Figure 1). In addition to this information gap, there are currently no studies regarding breast milk composition from these women with altered lipids during lactation. 

In this concern, it is well described that breast milk lipids change with the lactation stage [173]. At the same time, considering the first 16–21 weeks postpartum, total breast milk lipids increase with the length of lactation. The changes include increased TG levels [174,175], a decrease in breast milk cholesterol levels [175,176,177], and so do free cholesterol levels [173]. On the other hand, women who exclusively breastfeed their infants excrete large amounts of cholesterol into their breast milk [178], which likely significantly decreases the serum cholesterol concentration in these women [178]. However, since lipids are not measured during gestation or postpartum if a nursing mother has increased the lipid levels, she might transfer them to her infant, likely impacting the offspring’s cardiovascular health early in life. 

Thus, there is a significant gap in information regarding the impact of dyslipidemia or hypercholesterolemia in women during the postpartum period and among those who breastfeed on their cardiovascular health, their infant, and breast milk composition. 

### 5.4. Hypertension

Hypertension, defined as persistently elevated blood pressure, is a significant risk factor for cardiovascular disease and affects approximately 1.28 billion adults worldwide, according to the WHO [179]. Globally, the prevalence of hypertension is slightly higher in men (24%) than in women (20%), though women are particularly vulnerable during pregnancy due to conditions like preeclampsia (PE) [179,180]. Hypertension affects about 5–10% of pregnancies and can lead to severe complications, including PE, preterm birth, and an increased risk of cardiovascular disease later in life for both mother and child [181,182]. Furthermore, studies indicate that 39% of women experience persistent hypertension at three months postpartum [183,184]. These statistics highlight the critical need for effective monitoring and management of blood pressure during and after pregnancy to reduce long-term health risks for women and their offspring.

It is well described that lactation between 1 and 6 months or higher is a protective risk factor for hypertension postpartum [185]. It is associated with a beneficial effect on maternal blood pressure that persists for decades, suggesting a protective association of lactation with maternal cardiovascular risk [186]. Unfortunately, though this protective effect was present in women who developed gestational hypertension, it was not among those who developed PE [187].

Among the women who developed hypertension during gestation, there was a delayed initiation of breast milk expression [188], significantly higher odds of reporting insufficient milk supply [189], increased odds of non-exclusive breastfeeding at four months postpartum [189,190] close to six weeks less than 12 months postpartum [189] Thus, they showed a shorter duration of lactation [190].

Regarding the nutritional composition of colostrum and mature breast milk from hypertensive women, they presented higher levels of total protein [191], higher amounts of lipids, calories [191,192], and carbohydrates in comparison to normotensive women [192]. Breast milk of mothers who had systemic arterial hypertension during pregnancy presented lower levels of fat and calories than women who did not have this condition [193]. On the other hand, in breast milk from women with PE, macronutrient composition did not present differences with respect to normotensive women during the lactation period [194].

Regarding breast milk metabolites in PE women compared to the control group at 6 months postpartum, low levels of glycerophosphocholine and citrate were described, both related to inflammatory responses and oxidative stress [195,196] and impaired energy metabolism [195,196].

Other compound concentrations, such as docosahexaenoic acid (DHA), were significantly increased in breast milk despite lower maternal plasma DHA concentrations in PE women [197], likely suggesting a local mammary gland regulation. Neurotrophins present in breast milk, such as milk nerve growth factor (NGF) and milk brain-derived neurotrophic factor (BDNF) levels, were lower at 1.5 months (10.5%) in the PE group as compared to the normotensive group [198], possibly causing the reported impaired growth parameters of children born to PE mothers.

As shown here, breast milk from mothers with hypertensive disorders presents alterations in some of its components (Figure 1). However, the impact of this dysregulation on the offspring has been neglected, evidencing a significant opportunity for further study.

### 5.5. Metabolic Syndrome

Metabolic syndrome (MetS) is a cluster of metabolic abnormalities, including hypertension, central obesity, insulin resistance, and atherogenic dyslipidemia, that significantly increase the risk of developing diabetes and cardiovascular disease [199]. According to global estimates, MetS affects approximately 20–25% of the adult population, with women having a significantly higher risk of prevalence compared to men [200,201]. During pregnancy, maternal MetS can increase the risk of gestational diabetes, preeclampsia, and adverse neonatal outcomes, including preterm birth and fetal overgrowth [202,203,204,205]. The prevalence of MetS at three years postpartum is 20% [206]. This highlights the importance of addressing metabolic risk factors early in pregnancy and continuing management into the postpartum period to improve long-term maternal and child health outcomes.

A meta-analysis conducted four years ago mentions that although most of the included studies revealed beneficial associations between lactation and maternal MetS development- the pooled effect among them was non-conclusive [207]. Notwithstanding, breastfeeding duration, among other factors, can predict MetS in lactating women [208,209].

The duration of breastfeeding was negatively associated with the risk of MetS incidence in women (hazard ratio (HR) 0.98, 95% CI 0.98–0.99). In particular, a one-month increase in breastfeeding duration reduced the risk of MetS by 2% [210].

This comparison was also made for women who presented or did not have GDM. The HR of MetS between GDM and non-GDM women demonstrated significantly reduced MetS incidence with a longer duration of exclusive BF (HR 0.93, 95% CI 0.88–0.98) [210]. Also, the impact of maternal obesity and obesogenic diets during lactation is the development of metabolic syndrome in adulthood [211].

Currently, there is no information on metabolic syndrome (MetS) and its impact on breast milk composition or quality, nor regarding the impact on the offspring’s health (Figure 1). Thus, the lack of studies in this field is an opportunity to understand if there is any alteration in the biofluid and infants.

## 6. Epigenetic Programming Effects of Breast Milk Extracellular Vesicles on Infant Metabolic and Immunological Development

Breast milk extracellular vesicles (EVs) play a crucial role in the epigenetic programming of infants by influencing gene expression and developmental pathways, including immunity, neural development, and metabolic regulation [212]. These vesicles are secreted by the mammary gland epithelium and act as molecular carriers, delivering bioactive components such as microRNAs (miRNAs), long non-coding RNAs (lncRNAs), small interfering RNAs (siRNAs), proteins, and lipids [213,214,215,216]. EVs survive the infant’s gastrointestinal environment and are absorbed by tissues, enabling cell-to-cell communication through systemic circulation and facilitating the epigenetic programming that modulates cellular functions critical for infant growth and long-term health [217,218,219]. For this reason, they are considered important signaling molecules between mother and child [215,220].

Human milk contains more than 1400 distinct miRNAs and is considered one of the biological fluids with the highest miRNA concentrations [213,220,221]. Many of these miRNAs are encapsulated within exosomes, which protect them from degradation and ensure their stability and biological activity [220]. Evidence suggests that miRNA concentrations vary throughout lactation, reflecting the dynamic nature of breast milk’s molecular profile and its adaptation to the infant’s changing nutritional and developmental needs [222,223,224,225].

Maternal conditions such as diabetes mellitus, overweight or obesity, and the diet that she consumes are known to trigger epigenetic changes in offspring and can significantly influence the miRNA composition of human milk [220]. In this regard, Mirza et al. investigated the differences in expression of human breast milk miRNAs between mothers with type 1 diabetes and healthy controls [226]. They reported nine differentially expressed miRNAs, six were upregulated: hsa-miR-4497, hsa-miR-1246, hsa-miR-133a-3p, hsa-miR-3178, hsa-miR-1290, and hsa-miR-320d, while the remaining three were downregulated: hsa-miR-518e-3p, hsa-miR-629-3p, hsa-miR-200c-5p [226]. Functional analysis revealed that these upregulated miRNAs are involved in key pathways such as FoxO signaling, immune modulation, and inflammation regulation [226]. Notably, hsa-miR-4497 and hsa-miR-3178 increased macrophage TNF-α production, suggesting their potential role in influencing immune responses in infants of mothers with type 1 diabetes [226].

Shah et al. found that immune-related miRNAs such as miR-148a, miR-30b, and miR-let-7a are found in lower concentrations in breast milk from mothers with gestational diabetes and overweight/obesity, suggesting a direct impact of maternal health on milk composition [227,228]. These miRNAs, critical for regulating metabolic and adipogenic pathways, were linked to changes in infant growth and body composition, with miRNA-148a negatively associated with infant weight and fat mass and miRNA-30b positively associated with these measures [227,228]. miR-148a is the precursor of miR-148a-3p, the most abundant miRNA in breast milk, known for its neuroprotective and neuroproliferative effects [229]. Consequently, a reduction in miR-148a may increase the risk of childhood obesity and negatively impact the neurological development of children of obese mothers [229]. These findings suggest that altered milk miRNA profiles may influence offspring’s metabolic programming, potentially increasing the risk of adiposity and metabolic diseases later in life. Similarly, women who consumed a high-fat diet showed increased miR-67 and miR-27 concentrations in milk fat globule, while animal models fed with an obesogenic diet exhibited higher concentrations of miR-222 and reduced miR-200 and miR-26, likely modulating infant metabolism to meet dietary patterns [230]. Thus, breast milk EVs are vital mediators of epigenetic regulation, linking maternal conditions, lifestyle, and diet to infant development, with long-term implications for reducing the risk of immune and metabolic disorders.

## 7. Projections

Breastfeeding is considered the gold standard for infant nutrition during the first months of life. The composition of breast milk has evolved to provide optimal nutrition, immune protection, and regulation of growth, development, and metabolism for infants. It is well-established that breastfeeding improves the quality of life for both mother and child. However, it’s crucial to consider that not all breast milk has the same quality. A significant contributing factor is the mother’s health status, which is often left aside. Chronic inflammatory maternal diseases can alter the composition of breast milk, potentially impacting the health of the nursling with long-term consequences. Therefore, more research is needed to understand how different maternal pathologies affect breast milk components and their potential implications for the offspring later in life. Despite these considerations, it is essential to mention that breastfeeding is consistently recommended over formula. An important aspect of this review is raising mothers’ awareness about the impact of their health status, which they can positively change through a healthier lifestyle.

## Figures and Tables

**Figure 1 nutrients-17-00387-f001:**
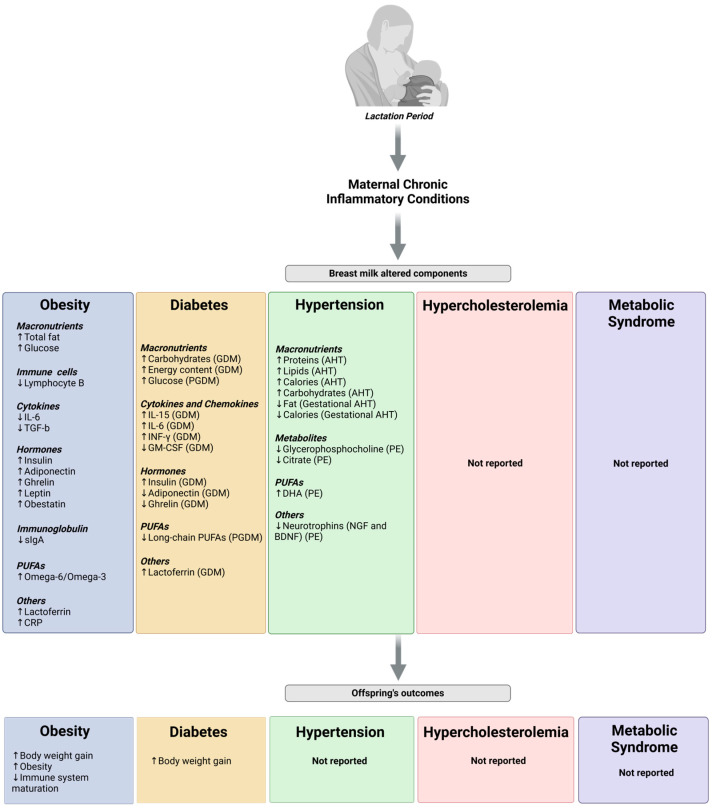
Diagram of altered breast milk components in women with chronic pathologies and the possible impact on the offspring. Abbreviations: CRP = C-reactive protein, GDM = gestational diabetes mellitus, PGDM = pregestational diabetes mellitus, AHT = arterial hypertension, PE = preeclampsia, PUFA = polyunsaturated fatty acids, NGF = nerve growth factor, BDNF = brain-derived neurotrophic factor. See text for references. Created with BioRender.com.

**Table 1 nutrients-17-00387-t001:** Maternal lipid-levels variation pre-pregnancy, during gestation, and postpartum period under diverse conditions.

Reference	Period, *n*, Variable	Maternal Lipid (mg/dL) Levels				
		TC	TG	LDL	HDL	Topic of the Study
Potter J et al., 1979 [136]	Nonpregnant, *n* = 15	202 ± 5	95 ± 10	129 ± 4	63 ± 4	Lipids levels during gestation and postpartum
	<8 weeks pregnancy, *n* = 6	200 ± 13	96 ± 12	-	-	
	9–12 weeks pregnancy, *n* = 5	198 ± 6	123 ± 13	-	-	
	13–18 weeks pregnancy, *n* = 9	223 ± 13	113 ± 9	137 ± 13	76 ± 4	
	19–24 weeks pregnancy, *n* = 10	279 ± 10	168 ± 21	179 ± 9	81 ± 5	
	25–30 weeks pregnancy, *n* = 13	274 ± 15	208 ± 18	176 ± 13	79 ± 11	
	31–36 weeks pregnancy, *n* = 23	297 ± 15	294 ± 24	189 ± 13	67 ± 4	
	37–40 weeks pregnancy, *n* = 25	317 ± 17	312 ± 23	214 ± 16	66 ± 3	
	Labor, *n* = 20	303 ± 17	306 ± 24	190 ± 14	75 ± 4	
	Delivery, *n* = 35	312 ± 14	296 ± 16	192 ± 11	82 ± 5	
	1 day postpartum, *n* = 23	268 ± 15	224 ± 19	168 ± 14	75 ± 4	
	5 days postpartum, *n* = 43	283 ± 11	174 ± 9	189 ± 11	73 ± 3	
	6 months postpartum, *n* = 30	257 ± 13	123 ± 8	184 ± 13	61 ± 4	
	12 months postpartum, *n* = 14	204 ± 12	117 ± 13	141 ± 11	52 ± 4	
Mizuno O et al., 1984 [155]	Pregnancy, *n* = 25	>300	-	-	-	Weight loss during postpartum and lipid levels
	Between 1 day and 8 weeks postpartum, *n* = 22	<250	-	-	-	
Montes A et al., 1984 [132]	Nonpregnant, *n* = 23	171 ± 26	59 ± 19	104 ± 23	56 ± 12	Lipids levels in gestation and postpartum
	34–38 weeks pregnancy, *n* = 23	251 ± 32	222 ± 60	161 ± 39	64 ± 9	
	6 weeks postpartum, *n* = 23	205 ± 23	71 ± 23	124 ± 21	64 ± 12	
	20 weeks postpartum, *n* = 23	190 ± 28	66 ± 18	120 ± 24	56 ± 11	
Knopp R et al., 1985 [162]	36 weeks pregnancy, *n* = 16, Antepartum no lactating	239 ± 41	228 ± 64	142 ± 35	64 ± 15	Lactation impact on maternal lipid level
	36 weeks pregnancy, *n* = 16, Postpartum no lactating	188 ± 29	92 ± 71	121 ± 30	51 ± 8	
	6 weeks postpartum, *n* = 16, Antepartum lactating	246 ± 44	221 ± 84	154 ± 39	68 ± 16	
	6 weeks postpartum, *n* = 16, Postpartum lactating	188 ± 29	92 ± 71	129 ± 31	65 ± 15	
Van Stiphout WA et al., 1987 [137]	Nonpregnant, *n* = 165	205 ± 3	-	-	57 ± 1	Serum lipid levels in young women before, during gestation, and in postpartum
	1 year before pregnancy, *n* = 29	~210	-	-	~60	
	Trimester 1, *n* = 16	~180	-	-	~60	
	Trimester 2, *n* = 17	~240	-	-	~70	
	Trimester 3, *n* = 20	~285	-	-	~70	
	1 year postpartum, *n* = 29	~200	-	-	~45	
Desoye G et al., 1987 [141]	Nonpregnant, *n* = 24	176 ± 40	87 ± 46	115 ± 41	44 ± 9	Hormones impact on lipid levels during pregnancy and postpartum
	8 weeks pregnancy, *n* = 42	167 ± 26	77 ± 28	98 ± 23	53 ± 13	
	38 weeks pregnancy, *n* = 42	286 ± 46	247 ± 84	165 ± 49	70 ± 18	
	6–8 weeks postpartum, *n* = 23, follicular phase of cycle	214 ± 32	123 ± 83	135 ± 30	52 ± 14	
	6–8 weeks postpartum, *n* = 23, luteal phase of cycle	214 ± 53	111 ± 110	142 ± 35	52 ± 15	
Jimenez D et al., 1988 [138]	12 weeks pregnancy, *n* = 60	182 ± 28	77 ± 21	105 ± 26	63 ± 13	Lipid changes during pregnancy and postpartum
	20 weeks pregnancy, *n* = 60	218 ± 42	106 ± 31	129 ± 36	69 ± 15	
	28 weeks pregnancy, *n* = 60	244 ± 47	143 ± 44	156 ± 43	66 ± 14	
	36 weeks pregnancy, *n* = 60	275 ± 44	180 ± 49	182 ± 42	65 ± 15	
	Delivery, *n* = 60	266 ± 68	192 ± 65	163 ± 64	64 ± 17	
	3 days postpartum, *n* = 60	225 ± 43	138 ± 43	142 ± 40	56 ± 13	
	40 days postpartum, *n* = 60	240 ± 46	80 ± 24	159 ± 42	64 ± 14	
Deslypere J et al., 1990 [160]	48 h postpartum, *n* = 209, parity 1	231 ± 45	175 ± 58	146 ± 43	54 ± 12	Impact of parity on maternal lipid levels
	48 h postpartum, *n* = 179, parity 2	248 ± 50	176 ± 50	163 ± 52	55 ± 13	
	48 h postpartum, *n* = 83, parity 3	245 ± 54	172 ± 56	160 ± 52	54 ± 13	
	48 h postpartum, *n* = 24, parity 4	252 ± 47	197 ± 81	173 ± 43	50 ± 13	
	48 h postpartum, *n* = 15, parity >5	251 ± 81	201 ± 38	168 ± 76	54 ± 13	
Loke D et al., 1991 [139]	Nonpregnant, *n* = 39	186 ± 29	74 ± 29	121 ± 29	56 ± 10	Lipids in pregnancy and postpartum under normal gestation
	28 weeks pregnancy, *n* = 67	257 ± 42	198 ± 60	152 ± 42	77 ± 14	
	32 weeks pregnancy, *n* = 67	257 ± 42	226 ± 74	154 ± 46	73 ± 14	
	6 weeks postpartum, *n* = 67	223 ± 43	122 ± 60	156 ± 45	52 ± 12	
Kallio M et al., 1992 [126]	2 months postpartum, *n* = 34	239 ± 58	72 ± 19	147 ± 37	62 ± 12	Lipids during and after prolonged lactation
	6 months postpartum, *n* = 28	201 ± 39	58 ± 13	120 ± 31	66 ± 13	
	9 months postpartum, *n* = 7	178 ± 15	50 ± 7	112 ± 16	58 ± 17	
	12 months postpartum, *n* = 8	182 ± 39	69 ± 25	112 ± 38	58 ± 13	
Chiang A et al., 1995 [140]	Nonpregnant, *n* = 184	~160	~90	~115	~40	Lipids during and after pregnancy
	Trimester 1, *n* = 62	~180	~100	~110	~50	
	Trimester 2, *n* = 62	~200	~110	~120	~50	
	Trimester 3, *n* = 62	~240	~230	~175	~55	
	Delivery, *n* = 62	~250	~220	~160	~55	
	6–12 weeks postpartum, *n* = 62	~180	~120	~130	~50	
Koukkou E et al., 1996 [142]	Trimester 2, *n* = 22, Normal glucose tolerance	260 ± 39	186 ± 189	155 ± 43	67 ± 12	Glucose metabolism alterations
	6–12 months postpartum, *n* = 22, Normal glucose tolerance	171 ± 31	83 ± 84	104 ± 23	67 ± 12	
	Trimester 2, *n* = 20, GDM	241 ± 62	256 ± 257	119 ± 46	66 ± 16	
	6–12 months postpartum, *n* = 20, GDM	179 ± 27	103 ± 106	105 ± 27	50 ± 12	
Van Dam R et al., 1999 [143]	Trimester 3, *n* = 266	~255	-	-	-	Women with or without depression
	4 weeks postpartum, *n* = 266	~232	-	-	-	
	10 weeks postpartum, *n* = 266	~209	-	-	-	
	16 weeks postpartum, *n* = 266	~201	-	-	-	
	22 weeks postpartum, *n* = 266	~195	-	-	-	
	28 weeks postpartum, *n* = 266	~193	-	-	-	
	34 weeks postpartum, *n* = 266	~190	-	-	-	
Troisi et al., 2002 [144]	Trimester 3, *n* = 46	291 ± 44	-	-	71 ± 18	Lipids levels and depression
	1–32 days postpartum, *n* = 45	235 ± 44	-	-	63 ± 14	
Rymer J et al., 2002 [166]	Trimester 3, *n* = 29	282 ± 57	313 ± 86	-	61 ± 15	Lipid levels in normal pregnancy
	12 weeks postpartum, *n* = 22	204 ± 34	144 ± 87	-	53 ± 19	
Lin C et al., 2005 [151]	6 weeks–2 years postpartum, *n* = 73, Normal glucose tolerance	203 ± 34	121 ± 68	-	57 ± 13	Glucose metabolism alteration
	6 weeks–2 years postpartum, *n* = 37, Abnormal glucose tolerance	212 ± 34	162 ± 123	-	52 ± 10	
	6 weeks–2 years postpartum, *n* = 17, Diabetes mellitus	213 ± 45	174 ± 185	-	56 ± 13	
Gunderson E et al., 2007 [123]	7 years postpartum, *n* = 48, No lactation	175 ± 32	64 ± 32	107 ± 29	55 ± 14	Lactation and weaning effect on maternal lipid levels
	10 years postpartum, *n* = 61, Post weaning	173 ± 30	61 ± 30	104 ± 28	57 ± 13	
Wiznitzer A et al., 2009 [145]	Nonpregnant 1 year, *n* = 3058	~170	~90	~100	~50	Lipid levels during pregnancy with preeclampsia and GDM
	Delivery, *n* = 3983	~240	~240	~140	~55	
	12 months postpartum, *n* = 2870	~160	~100	~100	~50	
Retnakaram R et al., 2010 [146]	Late trimester 2-early trimester 3, *n* = 87, normal glucose tolerance	~249	~230	~188	~63	Glucose metabolism alteration
	3 months postpartum, *n* = 87, normal glucose tolerance	~187	~70	~128	~57	
	late trimester 2-early trimester 3, *n* = 170, abnormal glucose tolerance	~244	~210	~181	~63	
	3 months postpartum, *n* = 170, abnormal glucose tolerance	~191	~81	~135	~55	
	late trimester 2-early trimester 3, *n* = 89, gestational impaired tolerance glucose	~239	~209	~174	~63	
	3 months postpartum, *n* = 89, gestational impaired tolerance glucose	~191	~91	~139	~50	
	Late trimester 2-early trimester 3, *n* = 136, GDM	~239	~220	~183	~59	
	3 months postpartum, *n* = 136, GDM	~205	~101	~149	~52	
Schwarz E et al., 2010 [163]	≥3 months postpartum, *n* = 121, Consistent lactation	~193	-	-	-	Lactation effect on maternal lipid levels
	<3 months postpartum, *n* = 84, inconsistent lactation	~192	-	-	-	
	postpartum, *n* = 92, no lactation	~197	-	-	-	
Prairie B et al., 2012 [147]	1–14 weeks postpartum, *n* = 120	196 ± 39	-	125 ± 34	50 ± 14	Maternal lipids levels and depression
Mendieta-Zerón H et al., 2013 [150]	3 days postpartum, *n* = 14, control	162 ± 4	228 ± 66	-	-	Preeclampsia impacto on maternal lipid levels
	3 days postpartum, *n* = 11, uncomplicated preeclampsia	207 ± 81	304 ± 162	-	-	
	3 days postpartum, *n* = 22, complicated preeclampsia	181 ± 54	244 ± 60	-	-	
Puhkala J et al., 2013 [156]	1 year postpartum, *n* = 464, B, Weight loss	182 ± 23	104 ± 33	85 ± 19	65 ± 12	Impact of weight on maternal lipid levels
	1 year postpartum, *n* = 464, FU, Weight loss	174 ± 32	75 ± 26	88 ± 22	59 ± 16	
	1 year postpartum, *n* = 464, B, No changes	178 ± 27	103 ± 53	85 ± 16	63 ± 12	
	1 year postpartum, *n* = 464, FU, No changes	186 ± 31	85 ± 44	97 ± 23	58 ± 11	
	1 year postpartum, *n* = 464, B, Weight gain	182 ± 27	109 ± 35	81 ± 19	67 ± 12	
	1 year postpartum, *n* = 464, FU, Weight gain	177 ± 30	96 ± 40	91 ± 21	55 ± 12	
Torris C et al., 2013 [164]	0–54 months postpartum, *n* = 43, <10 months lactation	185 ± 25	81± 43	-	58 ± 12	Impact of lactation duration on maternal lipid levels
	0–54 months postpartum, *n* = 55, >10 months lactation	167 ± 32	59 ± 20	-	57 ± 14	
Gaillard M et al., 2014 [161]	6 months postpartum, *n* = 4994, Parity 0	162 ± 23	~	93 ± 23	50 ± 12	Impact of parity on maternal lipid levels
	6 months postpartum, *n* = 2721, Parity 1	162 ± 23	~	93 ± 23	54 ± 12	
	6 months postpartum, *n* = 939, Parity 2	162 ± 23	~	89 ± 19	54 ± 12	
	6 months postpartum, *n* = 377, Parity ≥3	159 ± 23	~	89 ± 19	54 ± 12	
O’Higgins AC et al., 2017 [128]	6 weeks postpartum, *n* = 195, general population	~178	~92	~95	~64	Impact of GDM on maternal lipid levels
	6 weeks postpartum, *n* = 98, GDM	~198	~115	~117	~59	
Nouhjah S et al., 2017 [148]	6–12 weeks postpartum, *n* = 86, Control	173 ± 32	93 ± 54	104 ± 28	50 ± 11	Impact of GDM on maternal lipid levels
	6–12 weeks postpartum, *n* = 176, GDM	192 ± 48	116 ± 74	115 ± 31	52 ± 10	
Wahabi H et al., 2019 [157]	1 year postpartum, *n* = 34, Weight retention <3 Kg	166 ± 27	-	104 ± 23	-	Impact of weight retention on maternal lipid levels
	1 year postpartum, *n* = 40, Weight retention 3–7 Kg	162 ± 31	-	101 ± 27	-	
	1 year postpartum, *n* = 41, Weight retention ≥7 Kg	170 ± 31	-	108 ± 31	-	
Wen Ch et al., 2019 [149]	4 years postpartum, *n* = 25,558, normotension	172 ± 33	-	97 ± 29	57 ± 14	Impact of tension disorders on maternal lipid levels
	4 years postpartum, *n* = 1413, gestational hypertension	177 ± 36	-	103 ± 31	53 ± 14	
	4 years postpartum, *n* = 329, preeclampsia	183 ± 40	-	106 ± 35	53 ± 14	
Lim S et al., 2019 [167]	0–104 weeks postpartum, *n* = 74, B, >2 Kg of weight loss	201 ± 35	115 ± 62	124 ± 35	58 ± 12	Effect of weight on maternal lipid levels
	0–104 weeks postpartum, *n* = 74, 12 months after intervention, >2 Kg of weight loss	182 ± 35	97 ± 44	108 ± 31	54 ± 12	
	0–104 weeks postpartum, *n* = 74, B, weight stability ±2 Kg	197 ± 35	115 ± 71	120 ± 31	54 ± 16	
	0–104 weeks postpartum, *n* = 74, 12 months after intervention, weight stability ±2 Kg	186 ± 31	115 ± 53	112 ± 27	50 ± 16	
	0–104 weeks postpartum, *n* = 58, B, >2 Kg of weight gain	201 ± 39	106 ± 53	116 ± 35	58 ± 12	
	0–104 weeks postpartum, *n* = 58, 12 months after intervention, >2 Kg of weight gain	190 ± 31	124 ± 62	112 ± 31	50 ± 12	
Shalowitz M et al., 2019 [168]	6–12 months postpartum, *n* = 1029, race/ethnicity African	-	-	-	43 ± 15	Race/ethnicity on maternal lipid levels during postpartum
	6–12 months postpartum, *n* = 429, race/ethnicity Whites	-	-	-	43 ± 14	
	6–12 months postpartum, *n* = 450, race/ethnicity Latinas	-	-	-	45 ± 16	
Yang Z et al., 2021 [169]	30–330 days postpartum, *n* = 402	~175	~70	~130	~60	Chinese population
	12–16 weeks postpartum, *n* = 227, dyslipidemia	~259	~219	~145	~76	
Tinius R et al., 2021 [170]	Trimester 3, *n* = 25	-	208 ± 73	-	-	Metabolic changes from pregnancy to postpartum
	4–6 months postpartum, *n* = 25	-	65 ± 36	-	-	
Niu Z et al., 2022 [165]	<6 months postpartum, *n* = 28	165 ± 23	131 ± 58	102 ± 20	37 ± 12	Lipids levels and breastfeeding duration
	≥6–11 months postpartum, *n* = 24	170 ± 34	134 ± 81	103 ± 31	40 ± 10	
	12 months postpartum, *n* = 27	170 ± 32	90 ± 53	109 ± 24	44 ± 13	
Kyle E et al., 2022 [154]	4–6 weeks postpartum, *n* = 18, Physical activity and no lactation	163 ± 7	~96	92 ± 6.3	51 ± 4	Effect of physical activity and lactation on maternal lipid levels
	6 months postpartum, *n* = 18, Physical activity and no lactation	186 ± 8	~103	103 ± 7	62 ± 5	
	12 months postpartum, *n* = 18, Physical activity and no lactation	184 ± 8	~94	104 ± 9	59 ± 3	
	4–6 weeks postpartum, *n* = 18, Physical activity and lactation	172 ± 8	~76	106 ± 7	47 ± 3	
	6 months postpartum, *n* = 18, Physical activity and lactation	165 ± 6	~71	96 ± 5	54 ± 2	
	12 months postpartum, *n* = 18, Physical activity and lactation	157 ± 6	~65	90 ± 4	52 ± 2	
Hong K et al., 2023 [159]	Trimester 2, *n* = 869	243 ± 37	-	-	-	Weight and maternal lipids in postpartum
	Trimester 3, *n* = 869	270 ± 45	-	-	-	
	6 weeks postpartum, *n* = 869	208 ± 34	101 ± 66	128 ± 31	60 ± 13	
Zhang H et al., 2023 [158]	3–4 years postpartum, *n* = 315, <0 BMI points of change	178 ± 32	71 ± 126	101 ± 27	57 ± 12	BMI and maternal lipids in postpartum
	3–4 years postpartum, *n* = 671, 0–1.7 BMI points of change	176 ± 36	72 ± 66	101 ± 29	56 ± 12	
	3–4 years postpartum, *n* = 320, > 1.7 BMI points of change	183±34	92 ± 83	110 ± 29	52 ± 11	

Legend: ~ Symbol is used when the investigation does not mention an SD or SEM or when there is a graph, and the data are estimated from it. When the mean with range is given, only the mean was used. Abbreviations: *n* = number, B = baseline, FU = follow-up, BMI = body mass index, TC = total cholesterol, TG = triglycerides, LDL = low-density lipoprotein, HDL = high-density lipoprotein.

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
