# Peer review of "The Impact of Maternal Chronic Inflammatory Conditions on Breast Milk Composition: Possible Influence on Offspring Metabolic Programming"

_nutrients, 2025, doi:10.3390/nu17030387_

Round 1
Reviewer 1 Report
Comments and Suggestions for Authors
The article is highly comprehensive and thorough in its content. The English language is clear, the text is professionally accurate, and the terminology aligns well with the scientific context. The references are precisely integrated into the argumentation. The structure of the content is logical, and the subsections are distinctly separated. Have the authors investigated how maternal dietary habits (e.g., high-fat diets, etc) influence the lipid composition of breast milk? The article mentions that breastfeeding has a lipid-lowering effect but does not elaborate on how this might affect the mother’s long-term cardiovascular risk. Expanding on this aspect would be beneficial. The article highlights the role of microRNAs, but it is not entirely clear to what extent these are responsible for the long-term health outcomes of the offspring. Could more specific examples be provided in this regard? Is there any data available on how breast milk lipid and microRNA composition varies across different populations and geographic regions? The article is highly promising and valuable, and I recommend it for publication. The references (more than 190) are relevant, and the plagiarism index of the article is low.
Author Response
Answers to Revisor 1:
We are very grateful for your invaluable comments and suggestions, which make our work more understandable.
- Have the authors investigated how maternal dietary habits (e.g., high-fat diets, etc) influence the lipid composition of breast milk?
Thank you for your valuable suggestion. We have incorporated an extension to section 1, where lines 120 to 156 in the manuscript are exposed to what you asked for, with 24 new references.
- The article mentions that breastfeeding has a lipid-lowering effect but does not elaborate on how this might affect the mother’s long-term cardiovascular risk. Expanding on this aspect would be beneficial.
Thank you for the observation. A small paragraph regarding this information was added to the manuscript between lines 349 and 358.
- The article highlights the role of microRNAs, but it is not entirely clear to what extent these are responsible for the long-term health outcomes of the offspring. Could more specific examples be provided in this regard?
Thank you for the suggestion. Little information is available regarding the question, but one original study was included in the manuscript; please go to lines 545 to 548.
- Is there any data available on how breast milk lipid and microRNA composition varies across different populations and geographic regions?
Regarding this question, a paragraph was added in the first section of the manuscript between lines 157 to 189.
In total, 34 new references were added to the manuscript to resolve all the suggestions.
- Innis SM. Impact of maternal diet on human milk composition and neurological development of infants. Am J Clin Nutr. 2014 Mar;99(3):734S-741S.
- Lönnerdal B. Effects of Maternal Dietary Intake on Human Milk Composition. J Nutr. 1986 Apr;116(4):499–513.
- Bravi F, Wiens F, Decarli A, Dal Pont A, Agostoni C, Ferraroni M. Impact of maternal nutrition on breast-milk composition: a systematic review,. Am J Clin Nutr. 2016 Sep;104(3):646–62.
- Petersohn I, Hellinga AH, van Lee L, Keukens N, Bont L, Hettinga KA, et al. Maternal diet and human milk composition: an updated systematic review. Front Nutr. 2024 Jan 23;10.
- Insull W, Hirsch J, James T, Ahrens EH. THE FATTY ACIDS OF HUMAN MILK. II. ALTERATIONS PRODUCED BY MANIPULATION OF CALORIC BALANCE AND EXCHANGE OF DIETARY FATS*†. Journal of Clinical Investigation. 1959 Feb 1;38(2):443–50.
- KARMARKAR MG, RAJALAKSHMF R, RAMAKRISHNAN C V. Studies on Human Lactation. Acta Paediatr. 1963 Sep 21;52(5):473–80.
- Park Y, McGuire MK, Behr R, McGuire MA, Evans MA, Shultz TD. High‐fat dairy product consumption increases Δ9 c ’ 11 t −18∶2 (rumenic acid) and total lipid concentrations of human milk. Lipids. 1999 Jun;34(6):543–9.
- Mohammad MA, Sunehag AL, Haymond MW. Effect of dietary macronutrient composition under moderate hypocaloric intake on maternal adaptation during lactation. Am J Clin Nutr. 2009 Jun;89(6):1821–7.
- Ward E, Yang N, Muhlhausler BS, Leghi GE, Netting MJ, Elmes MJ, et al. Acute changes to breast milk composition following consumption of high‐fat and high‐sugar meals. Matern Child Nutr. 2021 Jul 3;17(3).
- Innis SM. Impact of maternal diet on human milk composition and neurological development of infants. Am J Clin Nutr. 2014 Mar;99(3):734S-741S.
- Kim H, Kang S, Jung BM, Yi H, Jung JA, Chang N. Breast milk fatty acid composition and fatty acid intake of lactating mothers in South Korea. British Journal of Nutrition. 2017 Feb 28;117(4):556–61.
- Liu Y, Liu X, Wang L. The investigation of fatty acid composition of breast milk and its relationship with dietary fatty acid intake in 5 regions of China. Medicine. 2019 Jun;98(24):e15855.
- Patterson E, Wall R, Fitzgerald GF, Ross RP, Stanton C. Health Implications of High Dietary Omega-6 Polyunsaturated Fatty Acids. J Nutr Metab. 2012;2012:1–16.
- Aumeistere L, Ciproviča I, Zavadska D, Andersons J, Volkovs V, Ceļmalniece K. Impact of Maternal Diet on Human Milk Composition Among Lactating Women in Latvia. Medicina (B Aires). 2019 May 20;55(5):173.
- Armand M, Bernard JY, Forhan A, Heude B, Charles MA, Annesi-Maesano I, et al. Maternal nutritional determinants of colostrum fatty acids in the EDEN mother-child cohort. Clinical Nutrition. 2018 Dec;37(6):2127–36.
- Mazurier E, Rigourd V, Perez P, Buffin R, Couedelo L, Vaysse C, et al. Effects of Maternal Supplementation With Omega-3 Precursors on Human Milk Composition. Journal of Human Lactation. 2017 May 23;33(2):319–28.
- Puca D, Estay P, Valenzuela C, Muñoz Y. Effect of omega-3 supplementation during pregnancy and lactation on the fatty acid composition of breast milk in the first months of life: a narrative review. Nutr Hosp. 2021;
- Ay E, Büyükuslu N, Batırel S, İlktaç HY, Garipağaoğlu M. The effects of maternal omega-3 fatty acid supplementation on breast milk fatty acid composition. ACTA Pharmaceutica Sciencia. 2018;56(2):27.
- Bortolozo EAFQ, Sauer E, Santos M da S, Baggio SR, Santos Junior G dos, Farago PV, et al. Supplementation with the omega-3 docosahexaenoic acid: influence on the lipid composition and fatty acid profile of human milk. Revista de Nutrição. 2013 Feb;26(1):27–36.
- Schwab U, Reynolds AN, Sallinen T, Rivellese AA, Risérus U. Dietary fat intakes and cardiovascular disease risk in adults with type 2 diabetes: a systematic review and meta-analysis. Eur J Nutr. 2021 Sep 21;60(6):3355–63.
- Xiao YL, Gong Y, Qi YJ, Shao ZM, Jiang YZ. Effects of dietary intervention on human diseases: molecular mechanisms and therapeutic potential. Signal Transduct Target Ther. 2024 Mar 11;9(1):59.
- Clemente-Suárez VJ, Beltrán-Velasco AI, Redondo-Flórez L, Martín-Rodríguez A, Tornero-Aguilera JF. Global Impacts of Western Diet and Its Effects on Metabolism and Health: A Narrative Review. Nutrients. 2023 Jun 14;15(12):2749.
- Kumar Verma M, Tripathi M, Kumar Singh B. Dietary Determinants of Metabolic Syndrome: Focus on the Obesity and Metabolic Dysfunction-Associated Steatotic Liver Disease (MASLD). In: Metabolic Syndrome - Lifestyle and Biological Risk Factors. IntechOpen; 2024.
- Monda A, de Stefano MI, Villano I, Allocca S, Casillo M, Messina A, et al. Ultra-Processed Food Intake and Increased Risk of Obesity: A Narrative Review. Foods. 2024 Aug 21;13(16):2627.
- Gunderson EP, Lewis CE, Wei GS, Whitmer RA, Quesenberry CP, Sidney S. Lactation and Changes in Maternal Metabolic Risk Factors. Obstetrics & Gynecology. 2007 Mar;109(3):729–38.
- Magnus MC, Wallace MK, Demirci JR, Catov JM, Schmella MJ, Fraser A. Breastfeeding and Later‐Life Cardiometabolic Health in Women With and Without Hypertensive Disorders of Pregnancy. J Am Heart Assoc. 2023 Mar 7;12(5).
- Perrine CG, Nelson JM, Corbelli J, Scanlon KS. Lactation and Maternal Cardio-Metabolic Health. Annu Rev Nutr. 2016 Jul 17;36(1):627–45.
- Kallio MJT, Siimes MA, Perheentupa J, Salmenperä L, Miettinen TA. Serum cholesterol and lipoprotein concentrations in mothers during and after prolonged exclusive lactation. Metabolism. 1992 Dec;41(12):1327–30.
- Shah KB, Fields DA, Pezant NP, Kharoud HK, Gulati S, Jacobs K, et al. Gestational Diabetes Mellitus Is Associated with Altered Abundance of Exosomal MicroRNAs in Human Milk. Clin Ther. 2022 Feb;44(2):172-185.e1.
- Shah KB, Chernausek SD, Garman LD, Pezant NP, Plows JF, Kharoud HK, et al. Human Milk Exosomal MicroRNA: Associations with Maternal Overweight/Obesity and Infant Body Composition at 1 Month of Life. Nutrients. 2021 Mar 27;13(4):1091.
- Słyk-Gulewska P, Kondracka A, Kwaśniewska A. MicroRNA as a new bioactive component in breast milk. Noncoding RNA Res. 2023 Dec;8(4):520–6.
- Zhang Z, Wang Y, Yang X, Cheng Y, Zhang H, Xu X, et al. Human Milk Lipid Profiles around the World: A Systematic Review and Meta-Analysis. Advances in Nutrition. 2022 Nov;13(6):2519–36.
- Kumar H, du Toit E, Kulkarni A, Aakko J, Linderborg KM, Zhang Y, et al. Distinct Patterns in Human Milk Microbiota and Fatty Acid Profiles Across Specific Geographic Locations. Front Microbiol. 2016 Oct 13;7.
- Yuhas R, Pramuk K, Lien EL. Human milk fatty acid composition from nine countries varies most in DHA. Lipids. 2006 Sep;41(9):851–8.
Reviewer 2 Report
Comments and Suggestions for Authors
The aim of the review is to determine the effects of inflammatory conditions on the composition of breast milk. The topic of the article is interesting but the way it is presented, the review has certain limitations. Given the nature of the review, it would be interesting for the authors to indicate the criteria they used to select the studies included in the review. On what basis do they consider the articles to be valid for inclusion? Since the composition of breast milk is discussed, the authors should include a brief section describing the composition of breast milk, otherwise there is a lack of context. In general terms, there is a lack of numerical data throughout the review. Likewise, the data obtained from these studies are not presented correctly. A better presentation in the form of Tables and Figures is needed. The only table present is difficult to interpret.
Other comments:
The type of studies studied for this review is not very well indicated in the abstract. Nor are the results obtained in this review indicated. Does it really affect? How does it affect? This is important because otherwise the abstract only presents what is going to be done but not the results and conclusions.
Line 26-37: This should really go in a methodology section after the introduction. Google Scholar is not a database, it is a search engine. The combinations of words used for the search are not clear, only single words are indicated. Please specify.
Line 128: Which pro-inflammatory omega 3 and omega 6 fatty acids?
In reality, as the Table is presented, it is very difficult to follow the data from the studies. With all the acronyms and the different studies, the differences are not very visual.
Author Response
Thank you for your valuable responses, including your comments and suggestions for making our revision more understandable.
In the following text are all the responses related to the observations done to our review.
- Given the nature of the review, it would be interesting for the authors to indicate the criteria they used to select the studies included in the review. On what basis do they consider the articles to be valid for inclusion?
Thank you for your comment:
Studies were included if they provided quantitative information on the relationship between previously mentioned variables and if the articles were written in English idiom. Please go to lines 62-64 in the manuscript.
- Since the composition of breast milk is discussed, the authors should include a brief section describing the composition of breast milk, otherwise there is a lack of context.
Regarding the suggested section on breast milk composition, you are right; thank you for your thoughtful comment. We have extended the first section, which previously, briefly, described this topic. You can find it in the manuscript between lines 96 to 119.
- In general terms, there is a lack of numerical data throughout the review. Likewise, the data obtained from these studies are not presented correctly.
Concerning numerical data, these data in the measure of the possible were added mainly in the lipid section, because this is the information available in the literature. See in the manuscript lines 96 to 156, 3741 to 379.
Comments 5 and 12 are similar, so they are answered together.
- A better presentation in the form of Tables and Figures is needed. The only table present is difficult to interpret/12. In reality, as the Table is presented, it is very difficult to follow the data from the studies. With all the acronyms and the different studies, the differences are not very visual.
The table we presented is now in a new version. We hope it is clearer and easier to interpret now. Please find it between lines 388 to 393.
The modifications done in the table are all highlighted in yellow for a better visualization.
Other comments:
- The type of studies studied for this review is not very well indicated in the abstract.
Line 25 in the abstract resolves this issue.
- Nor are the results obtained in this review indicated. Does it really affect? How does it affect? This is important because otherwise the abstract only presents what is going to be done but not the results and conclusions.
Thank you for your comment. Lines 21 to 23 in the abstract section clearly show what you asked.
- Line 26-37: This should really go in a methodology section after the introduction.
Thank you for your suggestion; a section on methodology was added to the manuscript; please go to lines 40 to 64 in the manuscript. In addition a small introduction was added as well, visit lines 29 to 38.
- Google Scholar is not a database, it is a search engine.
Google Scholar was corrected as a search engine as recommended.
- The combinations of words used for the search are not clear, only single words are indicated. Please specify.
Combination of words for the search was added to the methodology section as suggested, please go to lines 49 to 58 in the new version of the manuscript.
- Line 128: Which pro-inflammatory omega 3 and omega 6 fatty acids?
According to Panagos et al., there are no changes in omega 6 fatty acids, specifically linoleic acid, gamma linoleic acid and arachidonic acid. On the other hand, omega 3 fatty acids were reduced (ALA, EPA, DPA, DHA); this is the reason why the ratio is altered. Here, we attach the results of that article.
PUFA n-6 |
|||
18:2n-6 (linoleic acid) |
16.6 (3.4) |
17.0 (3.0) |
0.74 |
18:3n-6 (gamma-linolenic acid) |
0.16 (0.06) |
0.16 (0.07) |
0.91 |
20:4n-6 (AA) |
0.5 (0.08) |
0.5 (0.1) |
0.7 |
PUFA n-3 |
|||
18:3n-3 (ALA) |
1.7 (0.6) |
1.2 (0.3) |
0.008* |
20:5n-3 (EPA) |
0.8 (0.04) |
0.6 (0.03) |
0.013* |
22:5n-3 (DPA) |
0.2 (0.03) |
0.1 (0.05) |
0.0025* |
22:6n-3 (DHA) |
0.3 (0.1) |
0.2 (0.06) |
0.001* |
Regarding the other reference cited in our manuscript (Enstad S, 2021), concerning the altered ratio of omega 6/omega 3, this article measured polyunsaturated fatty acid, but it only declared the omega 6 and 3 ratios. Thus, they did not give the table with the fatty acid profile, and thus a specific detail of what is changing in breast milk in this regard. Therefore, we cannot tell if the alteration in the ratio is due to an increase in omega 6 or a decrease in omega 3 lipids in this specific article.
Round 2
Reviewer 2 Report
Comments and Suggestions for Authors
The limitations of the article are the same as before. That said, the authors have made an effort to improve the article.